# Nonclassical mechanisms to irreversibly suppress β-hematin crystal growth

Wenchuan Ma[1,5], Victoria A. Balta [2,5], Weichun Pan [1,3], Jeffrey D. Rimer [1,4✉], David J. Sullivan [2✉] & Peter G. Vekilov [1,4✉]

Hematin crystallization is an essential element of heme detoxification of malaria parasites and its inhibition by antimalarial drugs is a common treatment avenue. We demonstrate at biomimetic conditions in vitro irreversible inhibition of hematin crystal growth due to distinct cooperative mechanisms that activate at high crystallization driving forces. The evolution of crystal shape after limited-time exposure to both artemisinin metabolites and quinoline-class antimalarials indicates that crystal growth remains suppressed after the artemisinin metabolites and the drugs are purged from the solution. Treating malaria parasites with the same agents reveals that three- and six-hour inhibitor pulses inhibit parasite growth with efficacy comparable to that of inhibitor exposure during the entire parasite lifetime. Time-resolved in situ atomic force microscopy (AFM), complemented by light scattering, reveals two molecular-level mechanisms of inhibitor action that prevent β-hematin growth recovery. Hematin adducts of artemisinins incite copious nucleation of nonextendable nanocrystals, which incorporate into larger growing crystals, whereas pyronaridine, a quinoline-class drug, promotes step bunches, which evolve to engender abundant dislocations. Both incorporated crystals and dislocations are known to induce lattice strain, which persists and permanently impedes crystal growth. Nucleation, step bunching, and other cooperative behaviors can be amplified or curtailed as means to control crystal sizes, size distributions, aspect ratios, and other properties essential for numerous fields that rely on crystalline materials.

[1] William A. Brookshire Department of Chemical and Biomolecular Engineering, University of Houston, Houston, TX 77204, USA. [2] W. Harry Feinstone Department of Molecular Microbiology and Immunology, Malaria Research Institute, Johns Hopkins Bloomberg School of Public Health, Baltimore, MD 21205, USA. [3] Department of Applied Chemistry, Zhejiang Gongshang University, Hangzhou, Zhejiang 314423, China. [4] Department of Chemistry, University of Houston, Houston, TX 77204, USA. [5] These authors contributed equally: Wenchuan Ma, Victoria A. Balta. ✉email: jrimer@central.uh.edu; dsulliv7@jhmi.edu; vekilov@uh.edu

Hematin is the product of the oxidation of heme released in the digestive vacuole of malaria parasites as they metabolize hemoglobin[1]. To defend against hematin toxicity, the parasites sequester it into the innocuous crystalline hemozoin[2]. Several antimalarial compounds, such as quinoline-class antimalarials[3,4] and the hematin adducts of artemisinin-class drugs, produced by the parasite metabolism[5,6], kill the parasites by inhibiting hematin crystallization, which boosts the concentration of toxic free hematin. To model how drugs with limited residence times clear *P. falciparum* parasites, we probe whether antimalarials may adopt pathways that lead to inhibition of hematin crystal growth that lasts after an inhibitor has been removed from the system. We test whether irreversible inhibition of hematin crystallization correlates with the effective suppression of malaria parasites by limited-time pulses of five antimalarial compounds.

We explore the reversibility of inhibition of the growth of β-hematin crystals (Fig. 1a), a synthetic analog to hemozoin[1]. To promote the physiological relevance of the obtained results, we use as the solvent citric buffer-saturated octanol, an analog to the lipid sub-phase in the parasite digestive vacuole[7–10]. We probed the activity of three quinoline-class antimalarials, pyronaridine (PY), chloroquine (CQ), and mefloquine (MQ), and the hematin

adducts of two artemisinin-class drugs[11,12], artemisinin (H-ART) and artesunate (H-ARS)[5]. The quinolines inhibit hematin crystallization both in vivo and in vitro[13–15]. The hematin adducts of artemisinin-class drugs form as a product of the hemoglobin metabolism in the parasite digestive vacuole[6,16,17] and also suppress hematin crystallization[5].

The mechanistic detail that we put forth illuminates numerous observations of irreversible crystallization inhibition in nature, where intricate crystal architectures evolve shepherded by minority solution components[18,19]. Furthermore, industrial crystals are guided to preferred morphologies by specific modifiers[20–23]. The modifiers' activities are commonly ascribed to their adsorption to specific crystal surface sites[23–26] and the reversibility of adsorption inevitably predicts that growth fully recovers after the inhibitor is removed[26,27]. Multiple instances of permanently poisoned crystals[28] and terminal crystal sizes[29,30] contradict this prediction, stand out of the realm of the classical inhibition mechanisms, and have thus far remained elusive.

## Results and discussion
### Reversible and irreversible inhibition of bulk β-hematin growth.
β-hematin crystals typically display {100}, {010}, and {011} faces (Fig.1a). The {100} faces are the largest[31] and

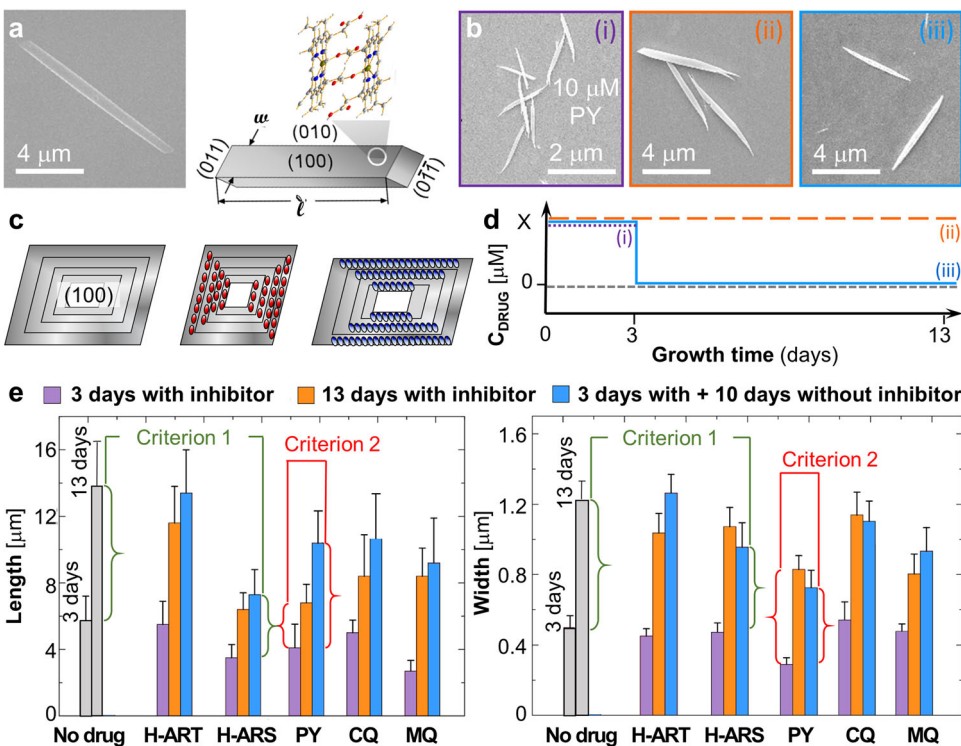

**Fig. 1 Inhibition of crystal length and width. a** A scanning electron microscopy (SEM) image and schematic illustrating the β-hematin crystal habit and the definitions of crystal length *l* and width *w*. In the hematin crystal structure, gray spheres represent C, blue, N, silver, H, and red, O. **b** SEM images of β-hematin crystals grown at times and compositions indicated with (i)–(iii) in (**d**) in the presence of 10 μM of PY. **c** Schematic of preservation of the crystal shape during growth in pure solutions and inhibitor-induced suppression of *l* or *w* by interaction of an inhibitor with axial and lateral crystal faces, respectively. Concentric contours denote crystal shape at increasing times of growth. **d** Schematic of inhibitor concentration variation used to test the reversibility of inhibition of *l* and *w*. Solid blue line indicates crystals exposed to an inhibitor concentration X μM (where X = 10 for H-ART, H-ARS, and PY; 2 for CQ; and 5 for MQ) for 3 days, after which the crystals were exposed to a metabolite- or drug-free solution for 10 days. The orange dashed line represents crystals that were kept at a constant inhibitor concentration X μM for 13 days. The dotted purple line indicates crystals exposed to inhibitor concentration X μM for 3 days; the short-dashed gray line indicates controls that grew in a metabolite- or drug-free solution for 13 days. Numbers (i)–(iii) indicate compositions and times of harvesting of crystals grown in the presence of PY and imaged in panel (**b**). **e** Lengths and widths of crystals grown in 0.5 mM hematin solutions and in the presence of 10 μM H-ART, H-ARS, and PY, 2 μM of CQ, and 5 μM of MQ in the starting solution. Error bars depict the standard deviations from averages over ca. 30 crystals for each composition regime and growth time. Vertical green and red braces define the length or width accrued during growth between days 3 and 13. Horizontal green and red brackets link length and width increments that are compared in, respectively, reversibility criterion 1, for H-ARS, and criterion 2, for PY.

commonly position parallel to a substrate[7]. With this orientation, the evolution of the crystal length, monitored by scanning electron microscopy (SEM, Fig. 1b), reveals the growth of the {011} faces, whereas the crystal width reports how fast the {010} faces grow (Fig. 1a, c). Notably, the distinct structures of the anisotropic crystal faces select distinct modes of inhibitor binding (to the kinks or on the terraces) and mechanisms and degrees of inhibition on each type of face. Thus, we do not expect a metabolite or drug to inhibit all faces uniformly, either reversibly or irreversibly. To assess whether the inhibition of the {011} and the {010} faces during bulk crystallization is reversible, we exposed β-hematin crystals to an inhibitor over 3 days and subsequently transferred them to drug-free hematin solutions for additional 10 days (solid blue line in Fig. 1d). The hematin concentration of the freshly prepared solutions was 0.5 mM, within the 0.2–0.6 mM range found in untreated parasites[6]. We compared the gains of lengths and widths of these crystals during the additional 10 days of metabolite- or drug-free growth to the equivalent gains of crystals that were exposed during both growth periods to either pure solutions (short-dashed gray line in Fig. 1d) or the same metabolite or drug concentrations (long-dashed orange line in Fig. 1d). The crystals from both reference groups were transferred into fresh solutions after the initial three days of growth. Irreversible inhibition would manifest as significantly shorter increases of length and width over 10 days of metabolite- or drug-free growth of crystals that were exposed to an inhibitor for initial three days compared to those crystals that were never exposed to an inhibitor (Criterion 1) and comparable to those of crystals that grew in the presence of an inhibitor for the entire 13 days (Criterion 2). Both criteria rely on length and width increments during growth in freshly added solutions, which minimizes biases due to potential drug-enforced delayed or enhanced nucleation and distinct crystal sizes that may enforce unequal supersaturation evolutions in the test and reference runs.

The lengths and widths of β-hematin crystals measured from SEM micrographs (Fig. 1e and Supplementary Fig. 1) indicate that H-ART appears to not affect either {010} or {011} faces. H-ARS and the quinoline-class drugs suppressed both the lengths and the widths of the crystals but did not completely stop growth, consistently with their known inhibition activities[15,26]. H-ARS inhibits the growth of both {011} and {010} faces irreversibly. The length increments for the last 10 days of crystals grown from pure solutions after exposure to H-ARS, ca. $4 \pm 1$ µm on average, are shorter than the ca. $8 \pm 2$ µm control growth exclusively in pure solutions (Criterion 1, Fig. 1e) and comparable to the length increments of the crystals that grew in H-ARS-rich solution for the entire 13-day period (Criterion 2, Fig. 2e). Similarly, the widths accrued during 10 days growth in a pure solution after three days growth in the presence of H-ARS are shorter than the equivalent width increments of crystals that grew both exclusively in a pure solution (Criterion 1) and in the presence of H-ARS for 13 days (Criterion 2). The parameters resulting from analysis of variance (ANOVA) of the distributions of the crystal lengths and widths (Supplementary Table 1) support the conclusions of reversible inhibition of the {011} and {010} faces by H-ART and irreversible inhibition by H-ARS of both {011} and {010} faces. Analogously, both criteria indicate irreversible inhibition of the crystal length and width by PY and of the crystal width by CQ (Fig. 1e and Supplementary Table 1). Irreversible inhibition of crystal length by CQ and MQ and of crystal width by MQ is implied only by Criterion 1 (Fig. 1e and Supplementary Table 1), suggesting a lower degree of irreversibility of the inhibition of the respective crystal faces by these two drugs.

**Molecular mechanism of irreversible inhibition.** The SEM observations of irreversible inhibition stand in sharp contrast to the current understanding of the molecular mechanisms of crystallization inhibition and how antimalarial agents affect hematin crystal growth. Recent work established that β-hematin crystals grow by nucleation of new layers, which spread by the incorporation of solute molecules in steps[7,32,33]. Atomic force microscopy (AFM) monitoring of the nucleation and growth of steps on β-hematin crystal surfaces at the lower hematin concentration of 0.28 mM (at the low end of the physiological range[6]) revealed that the antimalarial compounds selected for this study employ one of two classical mechanisms to inhibit β-hematin crystallization[5,15,26]. H-ART, H-ARS, and MQ adsorb at the kinks and partially block these unique sites where hematin molecules incorporate into steps[7]. PY and CQ follow an alternative inhibition mode, step pinning, whereby inhibitor molecules adsorb to flat terraces and arrest step growth over broad areas of the crystal surface[24,27]. The reversibility of adsorption assumed in the classical mechanisms[26,27] dictates that if an inhibitor is removed from the solution, it desorbs from the respective surface sites and step growth proceeds uninhibited[27]. The reversibility of β-hematin growth inhibition at a hematin concentration of 0.28 mM has been documented[5,7,15,26]. Step growth recovery may be delayed after the removal of inhibitors that adsorb strongly; growth resumes at its rate prior to the exposure to inhibitor when the inhibitors are buried in the crystal lattice[27]. If the solution concentration of an inhibitor is comparable to that of the solute, excessive inhibitor incorporation may strain the crystal lattice and impede crystal growth over longer times[28–30]. This latter mechanism, however, would not apply to the observed irreversible inhibition of β-hematin growth by H-ARS, PY, CQ, and MQ, whose concentrations, 10 µM and below, are at least 50-fold lower than that of hematin, 0.5 mM, at the upper end of the physiological range.

We explored by AFM whether irreversible inhibition of β-hematin crystallization may be due to unique behaviors triggered by the inhibitors at hematin concentrations elevated to 0.5 mM. We examined how hematin crystal surfaces and the solution, from which hematin crystals grow, respond to the introduction and removal of the five antimalarials at high supersaturation. At moderate hematin concentrations of 0.28 mM or lower, H-ART (Fig. 2a) acts exclusively as a kink blocker (Fig. 2b)[5]. In more concentrated 0.50 mM hematin solutions, however, time-resolved in situ AFM observations reveal that introduction of 10 µM H-ART—comparable to heme adduct concentrations found in trophozoites exposed to artemisinins[16]—causes the deposition of numerous hematin nanocrystals (identified from their characteristic habit, Fig. 1a) on the (100) β-hematin surface (Fig. 2c). In contrast to the nanocrystals that sporadically appear in inhibitor-free solutions at similar concentrations (Fig. 2d), the nanocrystals seen in the presence of H-ART do not grow (Fig. 2e). Consistently with the numerous nanocrystals observed by AFM, light scattering monitoring of 0.5 mM hematin solutions reveals a growing population of crystals and substantial acceleration of crystal nucleation in the presence of 10 µM H-ART (Fig. 2f–h). In both metabolite-free and H-ART-rich solutions AFM monitoring reveals decreasing trends of the step velocity $v$, the new layer nucleation rate $J_{2D}$, and the face normal growth rate $V$ as hematin concentration increases (Fig. 2i), an apparent violation of the law of mass action. In metabolite- or drug-free solutions the decreasing segments of the $v$, $J_{2D}$, and $V$ correlations with $c_H$ have been attributed to the shrinking of the terraces (Fig. 2b), which serve as essential conduits for the transfer of hematin molecules from the solution to the steps[33]. In the presence of H-ART, the decreasing trends start at much lower hematin concentration, where terraces are wide (Fig. 2c, e). We correlate this strongly counterintuitive behavior with escalating nucleation of nanocrystals that deposit on the surface

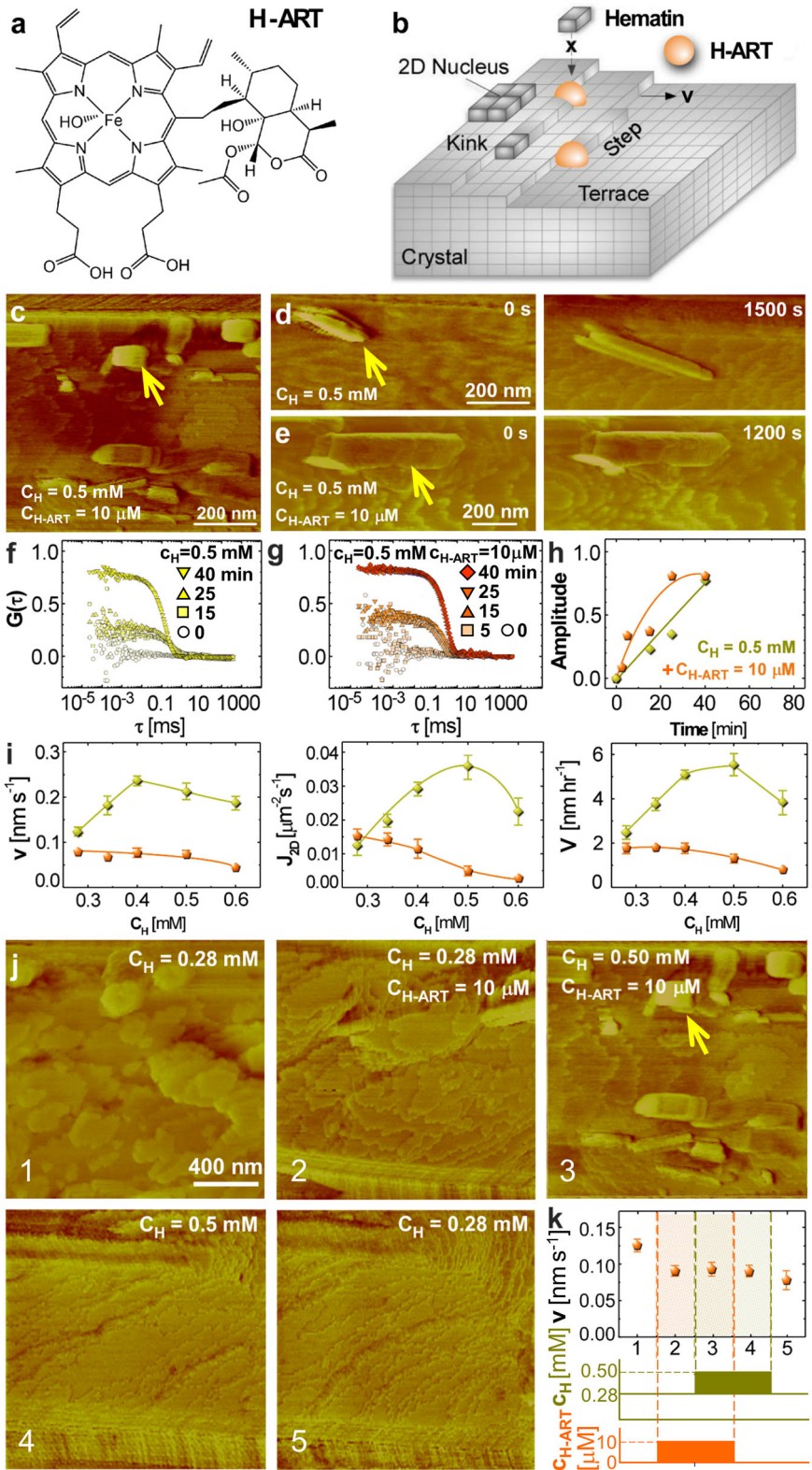

and, as the crystal grows, are buried in the crystal at random orientations (Fig. 2c, e, j).

The decreasing crystallization rates reveal that the buried crystals lower the crystallization driving force, likely by straining the crystal and increasing the chemical potential of the hematin molecules in the lattice[30,34,35]. To test whether the sequence of metabolite-driven nanocrystal nucleation, deposition, interment

in the large crystal, and higher lattice strain lead to irreversible inhibition, we cycled through two hematin concentrations, 0.28 and 0.5 mM, with either 0 or 10 μM H-ART (Fig. 2j, k). Removing H-ART arrests the nucleation of nanocrystals (Fig. 2j). The suspected lattice strain, however, persists after the metabolite is removed and suppresses β-hematin growth even in the absence of the inhibitor (Fig. 2j).

**Fig. 2 The molecular mechanism of irreversible inhibition of β-hematin crystallization by H-ART. a** The structure of H-ART. **b**, Schematic of step inhibition by kink blockers, which associate with the kinks and obstruct the access of solute molecules. **c–e** AFM images of (100) β-hematin crystal surfaces at hematin concentration $C_H = 0.50$ mM. Gold arrowheads indicate nanocrystals on β-hematin surfaces. **c** Still image with added H-ART at 10 µM. **d, e** The evolutions of nanocrystals in the absence of H-ART, in (**d**), and in its presence in (**e**). **f–h** Light scattering characterization of the evolution of aggregation in hematin solutions. **f, g** The evolutions of the autocorrelation functions $G(\tau)$ of the scattered light over 40 min in solutions with $C_H = 0.5$ mM, in (**f**), and with added 10 µM H-ART, in (**g**). Deviations from zero manifest the formation of aggregates. **h** The evolutions of the amplitudes of the aggregates' shoulders of $G(\tau)$ in (**f**) and (**g**), averaged over 10 measurements; the error bars denote the standard deviations and are smaller than the symbol sizes. **i** The correlations of step velocity $v$, the new layer nucleation rate $J_{2D}$, and the crystal growth rate $V$ with $C_H$. Gold diamonds, metabolite-free solutions; orange spheres, in the presence of 10 µM H-ART. Error bars denote the standard deviations from the averages of ca. 30 measurements. **j** Sequential AFM images of step patterns on the (100) β-hematin crystal surface at hematin and H-ART concentrations indicated on the panel. Gold arrowhead indicates a nanocrystal. **k** The step velocity $v$ at different combinations of hematin and H-ART concentrations, corresponding to morphologies in (**j**).

Similar to H-ART, adsorbed H-ARS reversibly blocks the access to kinks on β-hematin crystals growing at moderate hematin concentration[5]. When introduced to 0.5 mM hematin solutions, however, H-ARS elicits a response nearly identical to that of H-ART (Supplementary Fig. 2). H-ARS invokes abundant nucleation of nanocrystals, which incorporate into larger β-hematin crystals. The expected accumulation of lattice strain is the likely cause of the incomplete recovery of step growth after H-ARS is purged from the solution.

At moderate hematin concentrations, PY, (Fig. 3a) inhibits β-hematin growth by pinning the steps[15]. Step growth inhibition is amplified by the stabilization of step pairs (Fig. 3b), which evolve into step bunches and macrosteps[15]. PY did not boost the nucleation of nanocrystals and AFM did not show evidence of nanocrystals interfering with the growth of larger β-hematin crystals (Fig. 3c). PY exhibited an entirely different mechanism of irreversible growth suppression. At elevated solute concentrations, screw dislocations outcropped on the crystal surface (Fig. 3d) likely as a consequence of imperfect closure of solution inclusions engendered by the macrosteps[34] (Fig. 3d). The dislocations strain the lattice[34]; in addition, the high density of steps originating at the screw dislocations further delays step propagation, layer generation, and the crystal growth rate owing to the narrower terraces from which the steps feed[32,33] (Fig. 3e–g). Removal of PY does not restore the surface growth rate to its initial value since high densities of both steps and dislocations are preserved (Fig. 3h, i).

The response of the dynamics of the β-hematin crystal surfaces to CQ (Fig. 4a), a step pinner (Fig. 4b)[15], is different from that to H-ART, H-ARS, and PY. Unlike the two artemisinin adducts, an increase in hematin concentration in the presence of CQ does not lead to excessive nucleation of β-hematin nanocrystals in the growth medium (Fig. 4c). Moreover, increased supersaturation results in a higher step velocity (Fig. 4d, stage 3) and the removal of CQ from the growth solution results in a progressive increase in step velocity such that full recovery of surface growth rates to their initial state is observed within several minutes (Fig. 4d, stage 5). The recovery of the step velocity indicates that the action of CQ on the (100) face of β-hematin is fully reversible, in stark contrast to the unique behaviors of H-ART, H-ARS, and PY. Step generation and growth on the (100) face of β-hematin responds to high supersaturation and the presence of MQ, a kink blocker, analogously to the response to CQ (Supplementary Fig. 3). Notably, MQ and CQ do not boost the nucleation of hematin nanocrystals or the formations of step bunches and macrosteps.

Comparing the responses to kink blockers (H-ART, H-ARS, and MQ) to those to PY and CQ, two inhibitors that adsorb to the terraces, reveals that the reversibility of a modifier is not correlated to its mechanism of inhibition. Rather, inhibition may persist after the inhibitor is removed if it triggers cooperative responses, such as crystal nucleation/deposition or macrostep formation, that permanently impact the growing crystal in ways that suppress growth.

The molecular mechanisms activated by H-ART, H-ARS, and PY to irreversibly inhibit {100} faces of β-hematin suggest possible pathways that antimalarial agents may follow to inhibit the {011} and {010} faces. The nucleation of nanocrystals promoted by H-ARS and their out-of-registry incorporation into growing crystals is the likely cause of irreversible inhibition of the {011} and {010} faces. On the other hand, PY, CQ, and MQ do not enhance crystal nucleation, thus these antimalarials may irreversibly inhibit the {011} and {010} faces by interfering with the step dynamics, similar to the effects of PY on the {100} faces.

**Does irreversible inhibition of β-hematin crystallization relate to irreversible suppression of malaria parasites?** We tested whether irreversible inhibition of hematin crystallization correlates with inhibitor dose- and time-dependent suppression of malaria parasite growth. Notably, even if a metabolite or drug inhibits irreversibly only one of the hematin crystal faces (Fig. 1a), it will still delay the sequestration of hematin and contribute to the accumulation of the porphyrin, continuously produced by hemoglobin digestion. Thus, we expect the five compounds tested here to exhibit irreversible suppression of malaria parasites.

We chose the *P. falciparum* strain NF54, sensitive to most antimalarials, and carefully synchronized the ages of all parasites in the culture. We exposed the parasites to 3- or 6-h pulses of diverse drug concentrations, set at multiples of the half-maximal inhibitory concentration for a 72-h continuous exposure to a metabolite or drug (IC$_{50}$, Supplementary Fig. 4), after which we washed and incubated the parasites in metabolite- or drug-free media (Fig. 5a). We measured the fraction of surviving parasites using radioactive [³H] hypoxanthine, a biomarker for DNA replication. Hypoxanthine is a purine nucleobase that incorporates into newly-created DNA as a parasite begins its replication at the end of the second day of its erythrocyte lifecycle. The radioactive signal at the end of a 72-h incubation scales DNA replication with the fraction of surviving parasites[36,37].

To discriminate whether inhibitor activity relates to hemozoin suppression, we initiated the drug pulses at four parasite ages that differ by the stage of hemozoin crystallization (Fig. 5a): young rings, at which the hemozoin starts to nucleate, old rings, when the hemozoin crystals are small, and young and old trophozoites, at which stages in untreated parasites the hemozoin crystals grow[6]. All compounds, except PY, introduced to young rings and removed 6 h later inhibit substantially less parasite growth than when they were introduced at later parasite stages (Fig.5b). The weaker inhibitor activity at the parasite age when hemozoin crystals are few and small is consistent with hemozoin growth inhibition as the main pathway of inhibitor action. Furthermore, all five compounds partially protonate upon invading the digestive vacuole to adjust to its pH (ca. 5.0[38]) lower than that of blood and the erythrocyte cytosol (ca. 7.35) but undergo no further chemical modifications[13,17,39]. Thus, the weak response to inhibitor pulses applied to young rings suggests that the

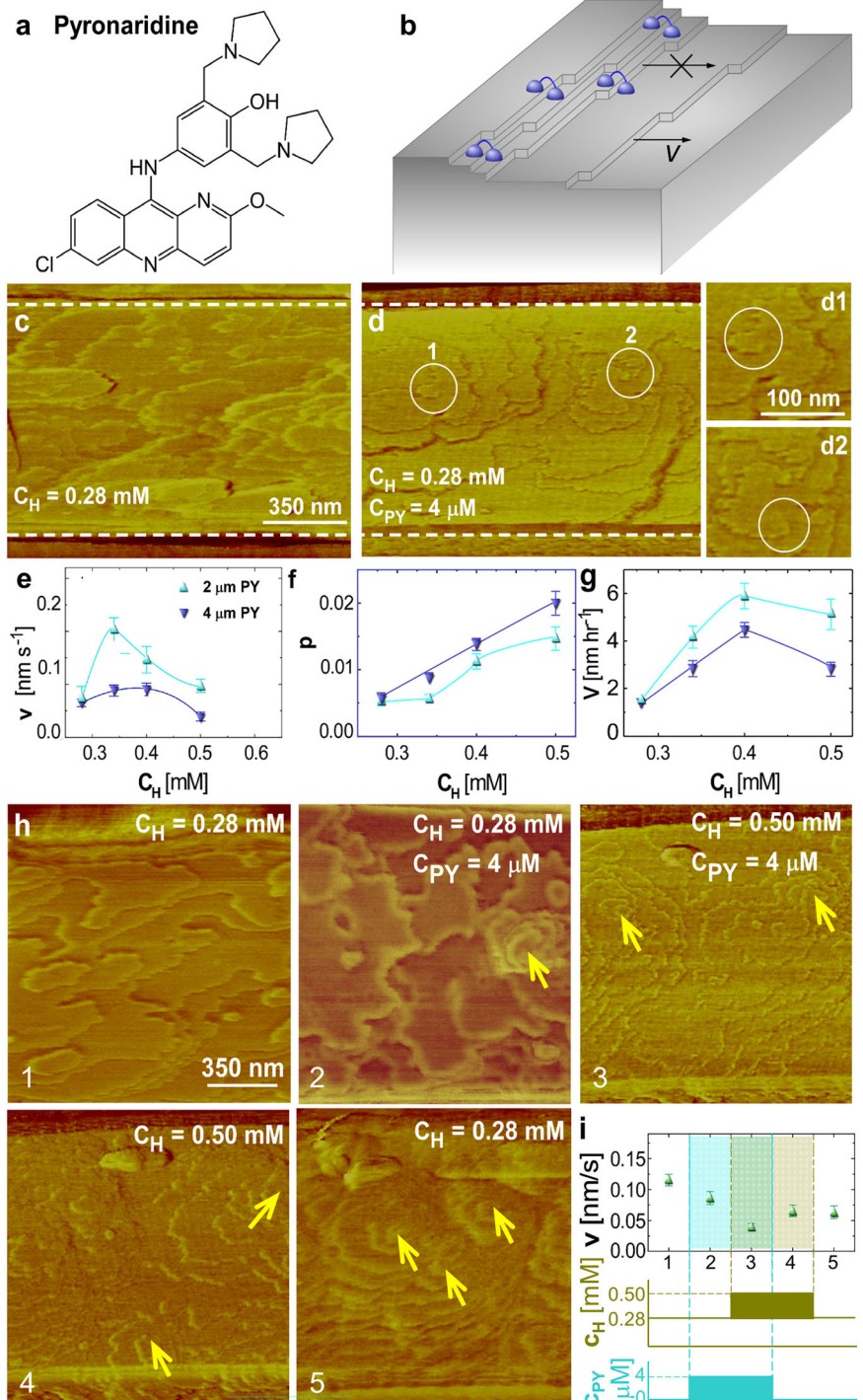

**Fig. 3 The molecular mechanism of irreversible inhibition of β-hematin crystallization by PY. a** The structure of PY. **b** Schematic illustration of step pair stabilization by PY. **c, d** AFM images of (100) β-hematin crystal surfaces at $C_H = 0.28$ mM, in (**c**), and after the addition of 4 μM PY, in (**d**). Callouts in **d** zoom in on encircled segments d1 and d2, where macrosteps fold to engender a dislocation. **e–g** The correlations of step velocity $v$, the dimensionless step density $p$, and the crystal growth rate $V$ with $C_H$ in the presence of 2 and 4 μM PY. Error bars span two standard deviations from the averages of ca. 30 measurements. **h** Step patterns on the (100) β-hematin crystal surface at hematin and PY concentrations indicated on the panel. Arrows indicate dislocation outcrop points manifested as centers of spiral steps. **i** The step velocity $v$ at different combinations of hematin and H-ART concentrations, corresponding to morphologies in (**h**).

compounds are not retained through later parasite stages and likely migrate out of the digestive vacuole.

Guided by the ratio of the inhibitor exposure times, 72 and 6 h, we define parasite suppression by an inhibitor as irreversible if the concentration at which a 6-h drug pulse inhibits parasite growth

by half is lower than 12-fold its $IC_{50}$ value, the concentration that inhibits 50% of parasite growth over 72-h of continuous drug exposure. Applying this criterion to the trophozoite survival fractions after 6-h pulses (Fig. 5b and Supplementary Table 2) reveals that the H-ART and H-ARS adducts required

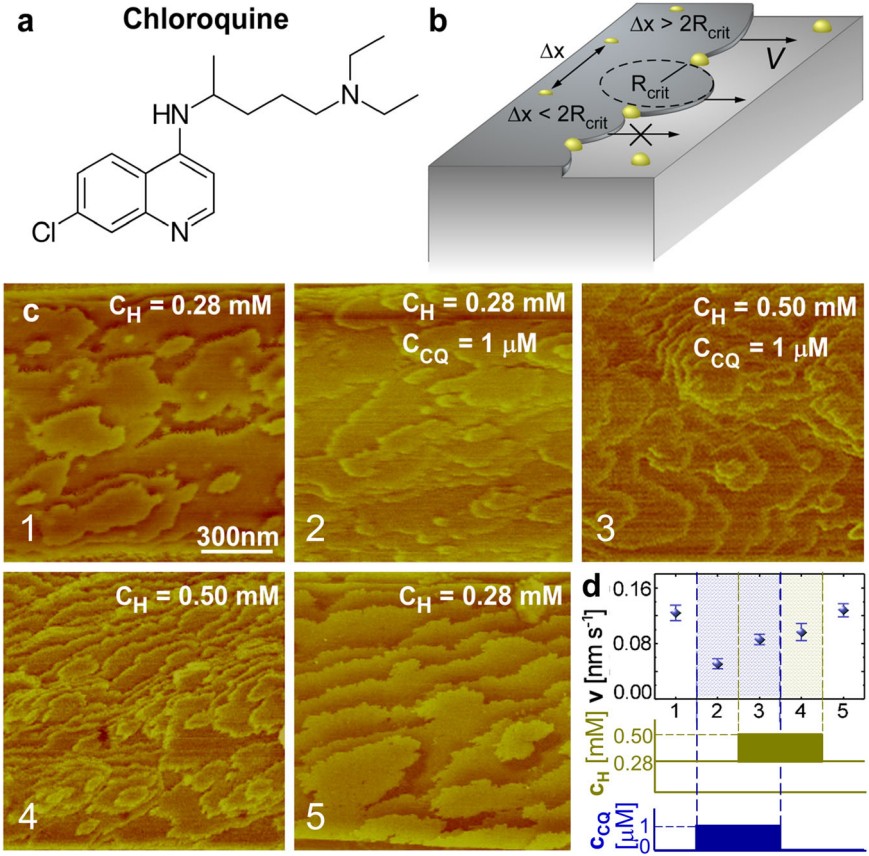

**Fig. 4 Reversible inhibition by CQ. a** The structure of CQ. **b** Schematic of step inhibition by step pinners, which adsorb on the terraces between steps, thus forcing steps to bend to grow between the two pinners. If the separation between two pinners $\Delta x$ is shorter than the critical two-dimension diameter $2R_{crit}$, step growth ceases. If $\Delta x$ is longer but comparable to $2R_{crit}$ step growth is delayed. **c** Step patterns on the (100) β-hematin crystal surface at hematin and CQ concentrations indicated on the panels. **d** The step velocity $v$ at different combinations of hematin and CQ concentrations, corresponding to the morphologies in (**c**).

$(5–11) \times IC_{50}$ for 50% trophozoite inhibition while the quinolines required ca. $(2–6) \times$ continuous $IC_{50}$. PYR achieved 50% ring stage pulse inhibition at $2 \times IC_{50}$ with CQ requiring $10 \times IC_{50}$ for ring stage inhibition, while MQ did not inhibit rings. CQ and H-ART had pulse drug $IC_{50}$ range for all stages in the fivefold to the ninefold range with a change in slope for young rings, H-ARS $10–11 \times$ for all but young rings at $17 \times$, MQ with trophozoite stages of twofold ratio with no inhibition of rings. MQ's ability to inhibit hemozoin crystals may be a secondary mechanism to protein synthesis or *P. falciparum* purine nucleoside phosphorylase inhibition[40–42]. PY had an IC50 ratio of $2 \times$ for rings, $5 \times$ for young trophs, and $9 \times$ for old trophozoites.

In general, all tested agents suppressed the parasite populations irreversibly when applied at the trophozoite stage or, in the case of PY, also at the early ring stage. PY pulses are most effective if applied to young rings, suggesting that PY may delay hemozoin crystal nucleation, consistent with the AFM observations of β-hematin crystallization (Fig. 3c, d, h).

Note that this pulse assay differs from the artemisinin ring stage pulse assays, which examine the activity of artemisinin drugs on ring stages from genetically diverse malaria parasites. Here we compared diverse compounds on a single *P. falciparum* isolate at four distinct stages.

The 3-h metabolite or drug pulses approximately doubled the IC50s and fold ratio except for PY which was approximately the same (Supplementary Fig. 5). The migration of the compounds out of the digestive vacuole, suggested by the weak responses of ring-stage parasites, signifies that the efficacy of short-term metabolite

or drug pulses is not due to residual inhibitor amount. Thus, the potent suppression by 3- and 6-h exposures to drugs is associated with irreversible suppression of hemozoin crystallization, which leads to hematin accumulation above the parasite's toxicity limit.

## Conclusions

We demonstrate that antimalarial agents mobilize two mechanisms to irreversibly suppress β-hematin growth even after the metabolite or drugs are removed from the growth solution. In both cases, irreversibility is likely enforced by strain on the crystal lattice. This strain emerges due to the cooperativity of molecules, which promotes nucleation of nanocrystals that incorporate into β-hematin crystals, or between growing steps, which bunch into macrosteps and produce abundant dislocations. Both cooperative modes of action are driven by high deviations from equilibrium. The newly identified correspondence between irreversible crystallization inhibition and irreversible suppression of malaria parasites suggests that irreversible inhibition of hematin crystallization may be essential for antimalarial' antiparasitic activity. In the context of crystal synthesis, these findings suggest that crystal sizes, size distributions, aspect ratios, and other properties essential for numerous fields may be controlled by amplifying or curtailing cooperative behaviors that invoke irreversible suppression of crystal growth and malaria parasites.

## Methods

**Materials**. The following compounds were purchased from Sigma Aldrich: hematin porcine (≥ 98%), citric acid (anhydrous, ≥ 99.5%), sodium hydroxide

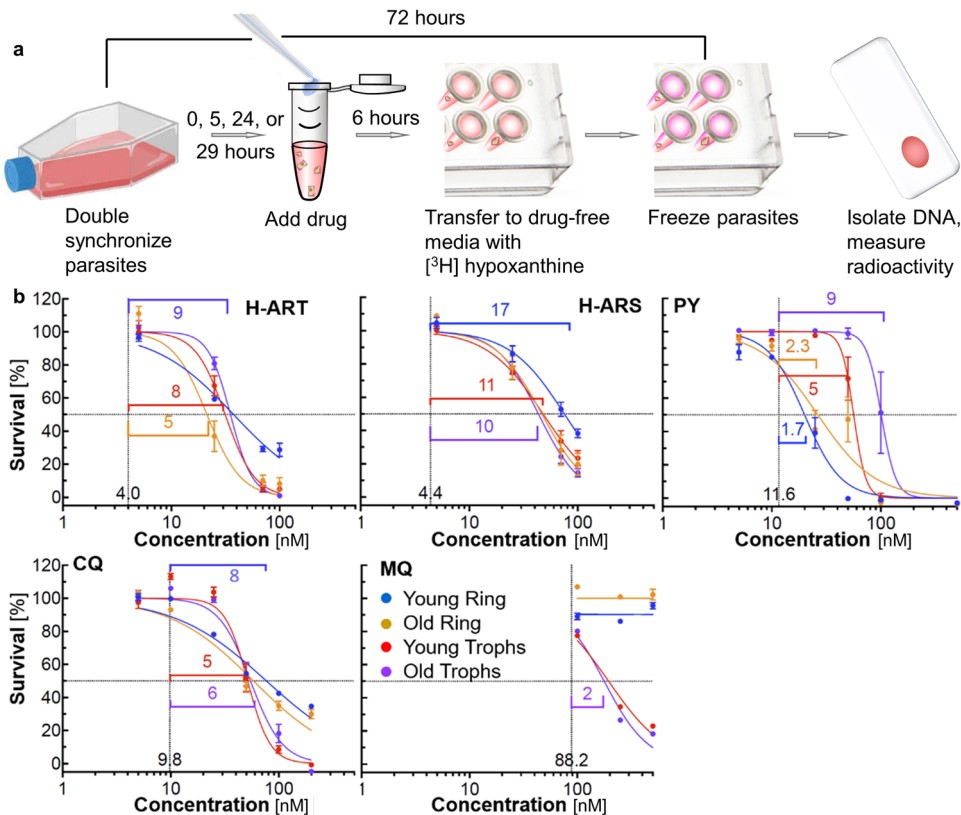

**Fig. 5 Irreversible suppression of malaria parasites in six-hour inhibitor pulses. a** Schematic of drug pulse assay with [3H] hypoxanthine detection of parasite survival. **b** Fraction of parasites surviving for 72 h after their lifecycles were synchronized. Concentrations refer to the inhibitor denoted in the respective panel. The parasites were exposed for 6 h to H-ART, H-ARS, PY, CQ, and MQ, introduced at 0, 5, 24, and 29 h of their lives. These parasite ages correspond to young ring-stage parasites, old ring-stage parasites, young trophozoites, and old trophozoites, respectively. The legend is in the MQ panel. Thin vertical lines and adjacent numbers denote independently measured continuous inhibitor $IC_{50}$ values, the concentration of a drug that inhibits 50% of the parasites with a 72-h inhibitor exposure. These horizontal lines mark the 50% parasite survival. Horizontal brackets and adjacent numbers denote fold ratios of the inhibitor concentrations that suppress 50% of parasites of each age in a 6-h pulse to the drug's continuous $IC_{50}$.

(anhydrous, ≥ 98%), *n*-octanol (anhydrous, ≥ 99%), porcine hematin, artemisinin (≥ 98%), artesunate (anhydrous, ≥ 98.0%), CQ diphosphate (≥ 98%), MQ hydrochloride (anhydrous, ≥ 98.0%). sorbitol, PY tetraphosphate (≥ 95%), hypoxanthine, saponin, sodium dodecylsulfate (SDS), bicarbonate, sodium bicarbonate (NaHCO₃, hypoxanthine, and HEPES. The following chemical were purchased from ThermoFisher Scientific: RPMI 1640 media supplemented with L-glutamine and gentamycin. Human serum was obtained from the interstate blood bank. All reagents were used as received without further purification unless otherwise noted. All materials were used as received. Deionized (DI) water was produced by a Millipore reverse osmosis ion exchange system (Rios-8 Proguard 2–MilliQ Q-guard).

**Crystal growth solution preparation**. Citric buffer at pH 4.8 was prepared by dissolving 50 mM of citric acid in DI water and titrating the solution, under continuous stirring, with the addition of 0.10 M NaOH to reach the desired pH as verified before each experiment using an Accumet Basic pH meter (Thermofisher Scientific). Fresh buffers were prepared every month and stored at ambient conditions. We placed 5 mL of citric buffer (pH 4.8) in direct contact with *n*-octanol at room temperature and allowed 30 minutes for equilibration. The upper portion of the two-phase system was decanted and denoted as citric buffer-saturated octanol (CBSO). Heme solutions were prepared by dissolving heme powder in 8 mL of freshly made CBSO and heating it to 70 °C for 7–9 h. The solution was filtered through a 0.2 μm nylon membrane filter (ThermoFisher Scientific) and the concentration was determined using a previously reported[7] extinction coefficient of $3.1 \pm 0.1\ cm^{-1}\ mM^{-1}$ measured at a wavelength of $\lambda = 594\ nm$.

**Synthesis of hematin adducts of artemisinin and artesunate**. Hematin solutions were prepared by a modified method using the same procedure as above but via the substitution of *n*-octanol with *n*-butanol. Sodium dithionite at ca. 1 mM and artemisinin (ART) at ca. 1 mM was dissolved in DI water and *n*-butanol, respectively. The heme solution was filtered with a 0.2 μm nylon membrane filter and placed in contact with the dithionite solution in a glass vial to yield a net molar ratio of 1:2:5 heme: ART: sodium dithionite. The vial was sealed under the flow of

nitrogen gas to create an inert atmosphere. The reaction involved the reduction of iron(III) to iron(II) in hematin with dithionite acting as the reducing agent. The system was maintained at 50 °C using a water bath (Super-Nuova Multi-Position Digital Stirring Hotplates). The aqueous and organic phases were rigorously mixed by shaking for ca. 30 s until the color of the solution changed from dark green to pink, indicating the reduction of hematin (containing Fe(III)) to heme (containing Fe(II)). The mixture was kept under static conditions for at least 30 min to allow for the separation of the organic and aqueous phases, after which the artemisinin solution was injected into the organic (top) fraction. The reaction between heme and artemisinin happened immediately after the injection, as judged by the instantaneous change in color from pink to orange. After allowing ca. 30 minutes for the reaction to reach completion, the organic layer was collected for later purification of the product, heme–artemisinin adduct (or H-ART).

The procedure to synthesize heme–artesunate adduct (H-ARS) was identical to that of H-ART with the replacement of ART with artesunate (ARS). The same reaction procedure was used with the only noticeable difference being a faster reaction to generate H-ARS, as gleaned by a more rapid color change following the addition of ARS to the heme(II) solution.

Further details about the procedures to purify and identify H-ART and H-ARS by mass spectroscopy are provided in Ref. [5].

**Preparation of β-hematin crystal substrates**. A hematin growth solution was prepared by dissolving hematin powder in 8 mL of freshly made CBSO followed by a 7–9 h period of heating at 70 °C. The solution was cooled to room temperature under ambient conditions and filtered with a 0.2 μm nylon membrane filter. The concentration was determined with an extinction coefficient $\varepsilon_{heme} = 3.1 \pm 0.1\ cm^{-1}\ mM^{-1}$ at $\lambda = 594\ nm$. The hematin solution was then diluted with fresh CBSO to achieve a final concentration of 0.20 μM. A piece of cover glass, cleaned with multiple water–ethanol–water cycles, was placed at the bottom of a glass vial and immersed in heme growth solution. Glass vials were sealed with closed-top septa caps and stored on a stationary platform in the dark at room temperature. Small crystals appeared on the glass slides after 2–3 days and reached a maximum size (ca. 20 μm) after two weeks. The glass slides containing crystals were rinsed with ethanol and DI water and dried in the air prior to analysis.

**In situ monitoring of crystal surface evolution**. Experiments were performed with a Multimode Nanoscope VIII and IV atomic force microscopes (AFM) from Digital Instruments. AFM images were collected in tapping mode (i.e., lightly engage) using Olympus TR800PSA probes (silicon nitride, Cr/Au coated 5/30, 0.15 N m$^{-1}$ spring constant) with a frequency of 32 kHz. Images were obtained using scan sizes of 0.3–20 μm, scan rates of 0.5–2.5 s$^{-1}$, 256 scan lines, and various scan angles depending on the orientation of the crystal substrate. The temperature in the liquid cell reached a steady value of 27.8 ± 0.1 °C within 15 min of imaging[15]. This value was higher than room temperature owing to heating by operation of the AFM scanner. The density of heme crystal substrates grown on glass disks (as described above) was monitored with an optical microscope to ensure an equivalent number of crystals for all samples (i.e., minimize potential depletion of free heme and growth inhibitor due to high total surface area of crystals). The glass slides were mounted on AFM sample disks (Ted Pella) and the samples were placed on the AFM scanner. Supersaturated heme solutions in CBSO were prepared less than 2 h in advance. The growth solution was loaded into the AFM liquid cell using a 1 mL disposable polypropylene syringe (Henck Sass Wolf), which is tolerant of organic solvents. After loading, the system was left standing for 10–20 min to thermally equilibrate. The crystal edges in optical micrographs were identified to determine the orientation and the crystallographic directions on the upward-facing (100) crystal surface. The crystals were kept in contact with the solution for 0.5–1.5 h to allow their surface features to adapt to the growth conditions.

The scan direction was set parallel to the [001] crystallographic direction and AFM images were collected for 3–5 h. The solution in the AFM liquid cell was exchanged every 30 minutes to maintain an approximately constant heme (an inhibitor) concentration. For studies of antimalarials, growth solutions were replaced with ones containing a selected inhibitor concentration. For each assay, the crystal substrates were first allowed to equilibrate (ca. 30 minutes) in growth solution without added inhibitor prior to the addition of solutions containing the inhibitor. For studies assessing irreversible inhibition, a series of solutions with varying heme and/or inhibitor concentrations were supplied to the AFM liquid cell at periodic imaging times. For all in situ measurements, the growth of heme crystal surfaces via two-dimensional (2D) layer generation and spreading was characterized by the velocity of step advancement $v$ (nm/s) and the rate of 2D nucleation of new crystal layers $J_{2D}$ (nm$^{-2}$ s$^{-1}$) using reported protocols[15]. In brief, we determine $v$ by monitoring the displacements of 8–13 individual steps with a measured step height $h = 1.17 \pm 0.07$ nm (corresponding to the unit cell dimension in the a-direction). Approximately, 25–35 measurements were taken for each individual step and the average step velocities were evaluated by analysis of sequential images over time. To determine $J_{2D}$, the appearance of new islands on the surface between successive images was monitored and the number of islands that grew was counted. This number was scaled with the scan area and the time interval between images to yield $J_{2D}$ (assessed from the average of 15–25 measurements).

Under most conditions tested here, multiple nuclei form at the same time and merge to cover the face. Under these conditions[43] the rate of crystal growth $V$ in direction normal to the observed face is evaluated as $V \cong \lambda^2 J_{2D} h \cong h(v^2 J_{2D})^{1/3}$ where $\lambda = (v/J_{2D})^{1/3}$ is the spacing between individual nuclei[33]. In the presence of PY hematin crystal surfaces are covered by trains of parallel steps. The nucleation of new layers cannot be observed and $J_{2D}$ cannot be measured. We measure the dimensionless step density $p = h/l$, whre $l$ is the separation between steps. The normal rate of growth $V$ is evaluated as $V = pv$.

**Light scattering characterization of β-hematin nucleation in hematin solutions**. Dynamic light scattering data was collected by an ALV instrument (ALV-GmbH, Germany), which includes ALV goniometer, a He-Ne laser with wavelength 632.8 nm, and an ALV-5000/EPP Multiple tau Digital Correlator. Normalized intensity correlation functions $G(q, \tau)$ were collected at a fixed scattering angle of 90° for 60 seconds. The characteristic diffusion time $\tau_c$ for hematin crystallites and the respective amplitude $A_c$ (which is proportional to the intensity scattered by the crystallites) were calculated by fitting $G(q, \tau)$ with an exponential relation[44,45] $G(q, \tau) - 1 = (A_c \exp(-\tau/\tau_c))^2 + \varepsilon(\tau)$, where $\varepsilon(\tau)$ is background noise in the correlation function.

**Tests for reversibility of inhibition of bulk hematin crystallization**. Hematin crystals were grown as described above at a concentration of 0.50 mM in the presence of an antimalarial drug. The inhibitor concentrations were concentration 10 μM for H-ART, H-ARS, and PY; 2 μM for CQ; and 5 μM for MQ. Three glass slides were inserted into the growth solution. After three days small β-hematin crystals were observed on the slides. One of the slides was transferred into a freshly prepared solution with the same components as the solution applied before and another one was transferred to a metabolite or drug-free solution with the same hematin concentration, 0.50 mM. These crystals grew for 10 more days. Crystals on the third glass slide were preserved as a reference. For control, crystals were grown for three days in the absence of any additives and then one of the slides was transferred to a fresh additive-free solution with the same initial hematin concentration. The slides with the crystals were extracted, dried, and imaged with SEM. For this, the slides were rinsed with DI water and dried with pure ethanol. The slides and the crystal attached to them were coated with 15 nm gold. The

widths and lengths of at least 30 crystals grown at each solution composition and inhibitor regime were measured and averaged.

To judge whether inhibition is reversible, we compared the increases in lengths and widths of these crystals during the additional 10 days of metabolite or drug-free growth to the equivalent increases of crystals that grew for 13 days either in pure solutions (short dashed gray line in Fig. 1d) or in the presence of the same inhibitor concentrations (long-dashed orange line in Fig. 1d). Irreversible inhibition would manifest as increases of length and width of the first group of crystals that are shorter than those of crystals that were never exposed to inhibitors (Criterion 1) and comparable to those of crystals that grew in the presence of inhibitors for 13 days (Criterion 2).

A greater number of crystals and larger crystal dimensions in one of the populations compared to Criteria 1 and 2 may empower faster solute consumption and lower supersaturation at longer growth times, leading to shorter lengths and widths that are not caused by inhibitor activity. Criterion 1 compares the average length and width increments of two crystal populations that grew, respectively, in the presence and absence of an inhibitor for the initial three days. The introduction of a fresh solution, in which the hematin concentration is equal to the initial 0.50 mM, minimizes the impact on the supersaturation evolution of a potentially unequal number of crystals due to inhibitor-induced faster or slower nucleation. The crystals that grew in the presence of an inhibitor for the initial three days were, on average, either about equisized or shorter and thinner than the crystals that grew in inhibitor-free solutions (Fig. 1e). Correspondingly, the solute consumption in the second solution would be similar to that in the vials with crystals that initially grew in inhibitor-free solutions or slower. The supersaturation at the later stages of growth would be equal to or higher than that in the vials with crystals that initially grew in inhibitor-free solutions. Thus, it appears unlikely that lower supersaturation during the secondary growth of crystals may delay the increase of crystal dimensions beyond the suppression due to irreversible inhibitor effects.

Criterion 2 compares the increases in the average length and width of crystal populations that were formed under identical conditions so that the numbers of crystals and the crystal dimensions are approximately equal at the beginning of the 10-day growth. The approximate equalities of crystal numbers and dimensions in these two populations ensure that during crystal growth the supersaturation stays approximately equal. Therefore, the observed differences in the average crystal lengths and widths are attributable to the irreversible inhibition by antimalarials.

**In vitro P. falciparum drug assay to test the reversibility of inhibition**. For this study, we obtained blood from living donors who are not identifiable to the researchers. Human erythrocytes are obtained from healthy volunteers under Johns Hopkins IRB-approved protocols.

CQ diphosphate (C-6628, Sigma Aldrich) and PY tetraphosphate (sc-205828A, Chem Cruz) were dissolved in sterile cell-culture grade H$_2$0. H-ART and H-ARS adducts, synthesized as discussed above, and MQ hydrochloride (M2319, Sigma Aldrich) were dissolved in DMSO (D128500, ThermoFisher). For use in parasite assays, we further diluted all tested compounds and adducts in inhibitor-free culture media ensuring DMSO or H$_2$O levels never reached above 1% final concentration when put into parasite culture.

In vitro experiments involved P. falciparum NF54 (MRA-1000) which were obtained through BEI Resources. Parasite cultures were maintained under a modification of the Trager and Jensen method[46]. Specifically, parasites were cultured at 2% hematocrit in RPMI 1640 media supplemented with L-glutamine, 25 mM HEPES, 0.25% NaHCO3, 0.37 mM hypoxanthine, 50 μL of 50 mg/ml gentamycin, and 10% human serum. Cultures were incubated at 37 °C in 5%CO$_2$/5%O$_2$/Balanced N$_2$.

To determine IC$_{50}$ for each inhibitor, P. falciparum cultures were synchronized to the ring stage by incubation in 5% sorbitol. Parasitemia was assessed by optical microscopy of a Giemsa-stained blood film. The IC$_{50}$ value was determined using a modified version of the [$^3$H]- hypoxanthine incorporation assay[37]. Each inhibitor concentration was performed in three technical replicates. Negative growth control wells contained 10 μM CQ. Positive growth control wells contained inhibitor-free culture media. Parasite cultures were plated in tissue culture-treated 96-well plates (353072, Falcon) at 2% hematocrit and 0.5% parasitemia in a final volume of 200 μl of hypoxanthine-free complete media. Parasite cultures were incubated with an inhibitor continuously for 72 h at 37 °C. At the time of incubation with inhibitor, 0.5 μCi of [$^3$H]-hypoxanthine was added to each well. Upon completion of the incubation period, 96-well plates were frozen at −80 °C until ready for sample harvesting. The 96-well plates were thawed, and samples were harvested onto glass fiber filters (GF/C, Brandel) by a cell harvester (MB48, Brandel). The incorporation of [$^3$H]-hypoxanthine was measured on a liquid scintillation counter. Parasite growth was determined by comparing the disintegrations per minute (DPMs) of control wells to test wells. IC$_{50}$ curves and values were generated using nonlinear regression analysis (log (inhibitor) vs. response) in GraphPad Prism 9 software (Extended Data Fig. 5).

P. falciparum cultures were synchronized to the ring stage by incubation in 5% sorbitol 48 h before, and immediately before, young ring pulse initiation. H-ART, H-ARS, CQ, PY, and MQ, were pulsed at concentrations from 0 to 500 nM on synchronized young ring stage cultures for 3 or 6 h, washed three times in culture media without hypoxanthine and then returned to complete culture media and incubated at 37 °C with 0.5 μCi of [$^3$H]-hypoxanthine added to each well. The old

ring pulse was initiated 5 h after the young ring pulse, the young trophozoite pulse was initiated 24 h after the young ring pulse, and the old trophozoite pulse was initiated 29 h after the young ring pulse. The full method is depicted graphically in Fig. 5a. Negative growth control wells contained 10 μM CQ. Positive growth control wells contained metabolite or drug-free culture media. Parasite growth was determined by comparing the DPMs of control wells to test wells.

**Statistics and reproducibility**. The average lengths and widths of crystals used to test the effects of drugs and metabolites on bulk crystallization (Fig. 1) were determined from measurements over 30 crystals.

We evaluate the step velocity *v* by monitoring the displacements of 8–13 individual steps. Approximately 25–35 measurements were taken for each individual step, for a total of ca. 300 measurements. The average step velocities were evaluated by analysis of sequential images over time.

To determine the rate of two-dimensional nucleation of new layers, $J_{2D}$, the appearance of new islands on the surface between successive images was monitored and the number of islands that grew was counted. This number was scaled with the scan area and the time interval between images to yield $J_{2D}$ from the average of 15–25 measurements.

In vitro *P. falciparum* experiments to test the reversibility of inhibition (Fig. 5) were performed in biologic duplicates on different dates with technical triplicate wells with noted drug dilutions. The Prism software package was used to generate dose–response vs inhibition with nonlinear fit.

**Reporting summary**. Further information on research design is available in the Nature Portfolio Reporting Summary linked to this article.

## Data availability
The datasets generated during and/or analyzed during the current study are available from the corresponding authors upon reasonable request. No custom-made computer code was used. Source data for figures can be found in Supplementary Data.

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

## Acknowledgements

This work was supported by NIH (Award no. R01 AI150763), NSF (Award no. DMR 2128121), and The Welch Foundation (Grants No. E-2170 and E-1794). D.J.S. acknowledges support from the Johns Hopkins Malaria Research Institute and The Bloomberg Family Foundation.

## Author contributions

W.M. and V.A.B. contributed equally. D.J.S. and P.G.V. conceived this project. P.G.V., D.J.S., and J.D.R. designed experiments. W.M. carried out all AFM and bulk crystallization experiments. W.P. carried out light scattering characterization. V.A.B. carried out all malaria parasite tests. W.M., V.A.B., J.D.R., D.J.S., and P.G.V. wrote the text.

## Competing interests

The authors declare no competing interests.
