## [Peer Review File · Communications Biology]

Reviewers' comments:

Reviewer #1 (Remarks to the Author):

REVIEWER REPORT 1

Dear Dr. Velikov, Dr. Sullivan and co-workers, artemisinins

The present manuscript describes a non-classical mechanism for the irreversible inhibition of β -hematin 2 crystal growth in presence of classical antimalarial quinoline and artemisin drugs. Within the findings, authors found that some antimalarial quinolines like PY, MQ, CQ and an artemisinin drug like the H-ARS were able to inhibit irreversibly the β -hematin crystal growth for sample of β -hematin crystal in absence of drugs but pre-treated with these drugs. Discrete or minimal changes were noted by using H-ART. These evidences were recognized by decrease in length and width size of β -hematin crystal concerning to untreated samples and, that phenomenon was noted for β -hematin crystal growth in presence and inclusive after removal of drug of the solution. It suggest that these drugs promoted irreversible changes in β -hematin crystal, which compromise its re-organization in absence of drug. Further experiments based on time-resolved in situ atomic force microscopy (AFM), complemented by light scattering, and reveals that quinoline drugs promote dislocations in crystals, which compromise the correct crystal growth through lattice strain way. It represents a step bunching, which affect the crystal integrity in terms of size, size distribution, radius and other key properties for a correct crystallization growth. These evidences were well correlated with biological experiments using different parasite at age-stages such as young ring-stage parasites, old ring-stage parasites, young trophozoites, and old trophozoites. Remarkable inhibition was recognized for parasite pre-treated and post-removal drug, which suggest that the in vitro chemistry finding can be extrapolated for the biological system. All these interesting offer an important contribution to the understanding of inhibition of β -hematin beyond classical mechanism based on presence of drug. Difference in non-classical irreversible inhibition between PY, MQ, CQ and H-ARS concerning H-ART represent an interesting result, which in my opinion could open way to future rational designs. The article offer an understanding of the described phenomenon from a molecular point of view. Thus, I recommend the present article for publication in Communication Biology. Only, I have some simple questions:

- (i) Beyond the difference found in AFM experiment between PY, MQ or CQ with H-ART, what structural aspects could be dominant in the discrete inhibition of H-ART after drug removal?
- (ii) Have author mediated the possibility to reply the biological studies by using CQ-resistant strain? It can be very interesting in order to explore the occurrence or nor of this non-classical mechanism in CQ-resistant strain of *P. falciparum*?
- (iii) In survival (%) vs. concentration plots of Figure 5, the units in x-axis are expressed as nm. Should be it mM?

With any other comment,

Best regards,

Reviewer 1

Reviewer #2 (Remarks to the Author):

COMMSBIO-22-3340-T

"Nonclassical mechanisms to irreversibly suppress β -hematin crystal growth"

Ma et al.

reviewers comments

In this study the authors form small beta hematin crystals (crystallized heme) in vitro to "seed" formation of larger beta hematin crystals from saturated heme solutions (identical to heme crystals

called hemozoin that are formed within growing malarial parasites) and monitor growth of those seed crystals in vitro for 3 vs 13 days (or 3 days followed by 10 days more) via atomic force microscopy in the presence vs absence of quinoline antimalarial drugs and artemisinin - heme adducts. They interpret their data in terms of two modes of crystal growth inhibition, either reversible or irreversible. The topic is quite important and the team of authors are prominent and highly skilled researchers, however, some points need to be addressed as described below.

Major issues:

1) The key data are presented in Fig. 1, and distinguishing whether blue and orange bars in Fig. 1e are the same or different in height is essential to interpretation, however, no statistics are given (also bars in extended data Fig. 2). From the error bars (extended Fig. 2 has no error bars at all) it is difficult to ascertain which blue and orange bars are statistically the same vs different. Routine T test and recitation of calculated p values is sorely needed.

2) The authors propose that inhibition of crystal growth is irreversible if two criteria are met, the first being less crystal growth vs control (which is no inhibitor for 13 days) in the 10 days after inhibitor is removed following a 3 day incubation with inhibitor (criterion 1) and, second, that growth in the constant presence of inhibitor for 13 days is the same as growth following 3 days + followed by 10 days - inhibitor. Only 1 inhibitor (H-ARS) seems to satisfy both, but the title and text of the paper seems to imply that the inhibitors are irreversible, and Fig. 2 presents detailed arguments and a cartoon entitled "irreversible inhibition ... by H-ART" when the data in Fig. 1 show that H-ART satisfies neither criterion. The separation of crystal width vs length effects as reversible or irreversible in the text further confuses interpretation, are the authors implying that some inhibitors are reversible in one dimension but irreversible in another ? Some clarification in simple language is needed to assist the reader.

3) The crystal inhibition assays are done over 13 days in vitro under highly non physiologic conditions, but the authors imply that by comparing growth effects of live parasites vs the same inhibitors that somehow the measured crystal inhibition characteristics are relevant to understanding the mechanism of drug inhibition of hemozoin in vivo, in which hemozoin crystals are formed within hours. The highly artificial nature of the crystal formation measurements needs to be emphasized, with limitations on interpretation then highlighted.

Additional points:

4) pg. 1 "inevitably predicts" but then the next sentence seems to contradict this.

5) next line, "irreversible inhibition of hematin crystallization" ... at best, "in vitro under non physiological conditions".

6) next line following, "cooperative", how so ? This term has a formal definition in biological sciences, it is not clear how cooperativity in inhibition of crystal growth by any inhibitor studied is being ascertained or quantified.

7) bottom of pg 1, top pg 2 is very misleading, H-ART and H-ARS used in the paper are not drugs, they are drug heme adducts. ART drugs cannot be "purged" from the solution as implied, they become covalently attached to their intracellular targets.

8) 5 lines following, "copious nucleation" is not defined or quantified, what is meant by this phrase ?

9) pg 2 par 2 last line, reference 23 does not suggest "adducts ... form in the ... digestive vacuole" as implied, this paper uses NMR methods to assign meso carbon covalent attachment sites for ART - heme adducts formed in vitro.

10) pg 12 last line second par, ref 32 measures killing rates which is not what is being measured here, but it is implied that what ref 32 and the authors measure is similar.

Methods

11) "Materials"; were any of the chemicals or drugs purified or were they used as purchased ?

12) "Synthesis of Hematin" Sodium dithionite and artemisinin ... concentrations are not mentioned

13) "In Situ Monitoring ..." Please describe "... the liquid cell ..." is this a commercial or fabricated device, etc.

14) Top pg. 3 *how* was the solution in the fluid cell "...exchanged ..." and is the fluid cell the same as the liquid cell mentioned earlier ?

15) "Tests for Reversibility ..." 3rd line, H-ART and H-ARS are not drugs. Next line, why 2uM and 5uM for CQ and MQ ? Without explanation this seems arbitrary.

16) pg 5 last par, drug and inhibitor concentrations are not listed. Also, "... parasite survival" is not the inverse of percent growth inhibition, but the inverse of parasite growth ?

Reviewer #3 (Remarks to the Author):

Heme detoxification suppression still stands as a pivotal treatment for malaria. Basic science towards understanding the nucleation of β -hematin crystals in the growth/biogenic medium and how antimalarial drugs can impair it is of importance. The manuscript characterizes the β -hematin growth under antimalarial drugs (old and new ones) towards advancing the underlying mechanisms of inhibition. A key advance here is to address each contribution of reversible and irreversible inhibitory steps. Notably, authors have identified irreversible inhibition of β -hematin crystallization by some drugs, and importantly, aimed to understand the underlying reason for the phenomena and to correlate this with the antiplasmodium activity. Overall, though there are many questions remaining about how exactly these compounds exert heme toxification and antiparasitic effects, there is sufficient new insights to support its publication in Communications Biology. That said, there are some important concerns outlined below should be addressed.

A) The notion that in low concentration, hematin-artemisinin adducts (H-ARS/H-ART) did not efficiently inhibit the growth of young parasites (early or late rings) is a substantial phenotype shift and a novelty in comparison to the parental Artemisinin/Artesunate efficacy. Presumably, adducts are devoid in peroxide bond necessary to alkylate protein; however, adducts were previously able, at least in a high concentration (500 nM), to kill early rings in the RSA (DOI 10.1074/jbc.RA120.016115). To reconcile this apparent paradox, authors are encouraged to discuss this or experimentally address the parasite survival in Figure 5 panels B and C using high concentrations of H-ARS/H-ART (up to 1 microM).

B) Yet regarding the drug concentration, the authors report IC50 values of 6 h expose versus 72 h (no

wash out). The determination of IC₅₀s 6h was presumably intended to reflect the irreversibility of parasite inhibition and to precisely correlate this phenomenon with the drug interaction with hemozoin crystals; this is a novelty. A cut-off of IC₅₀ 6 h > 12 folds the IC₅₀ 72 h was wisely established. Troublingly, the curves of IC₅₀ values of 6 h expose for early and later ring do not seem like a sigmoid curve. The accuracy of regression-derived values is not clear either. In other words, why authors did not test compounds in higher concentration up to 1 microM in order to generate suitable sigmoid curves?

C) In parts, a great novelty of the present study is the determination of phenotype signature of hematin-artemisinin adducts (H-ARS/H-ART). Their antiplasmodium activity are quite appealing. IC₅₀ values determined within 72 h of continuous drug incubation indicate they are almost equipotent, this is consistent to the structure-activity relationship. That said, there are dissimilarities in the IC₅₀ 6-h that should be addressed/discussed. There are a couple concerns with this experiment, though. First, H-ART seems to kill trophozoites more efficiently than H-ARS. Even in higher concentration, there are still parasites surviving at H-ARS treatment. Subsequently, trophozoite survival fractions for H-ART is of 5, while of 11 for H-ARS. Does this behave in the same way for IC₅₀ 3-h (Extended Data Fig. 5 not depicted for H-ARS)? Conversely, we know that iron protoporphyrin IX (Fe-PPIX) can adsorb in wire glass and the plastic surface of a microplate. No evidence is provided to indicate that the H-ARS/H-ART can be truly washed out by the protocol used. Therefore, it is possible that parasites continue to be effectively exposed to the hematin-artemisinin adducts following the washout step, especially if no plate transfer was performed (see plate transfer, DOI: 10.1128/AAC.00574-16). This could be the reason for the dissimilarity in IC₅₀ 6h values. This could be a useful feature in therapeutic applications, but confounds interpretation of phenotype response.

D) In a close inspection of the IC₅₀ values from Extended Data Fig. 6, all drugs except for mefloquine were quite consistent to the literature. For mefloquine, the IC₅₀ of 88 nM is higher than typically observed in most the literature. Mefloquine supplied by Sigma-Aldrich (M2319) is provided as a partially DMSO-insoluble salt. Could this be the reason for the limited potency of mefloquine?

E) Yet regarding mefloquine. The irreversibility of parasite inhibition and the correlation with the drug interaction with hemozoin crystals (reversible/irreversible) is an important issue. For sure, all drugs tested here apart mefloquine (chloroquine, pyronaridine, and hematin-adducts) are of fast-action antimalarial activity (i.e., to decrease parasite viability over 24 h drug expose). Presumably, a fast-action property may correlate with the ability of a drug in inhibiting hemozoin crystal. It is largely assumed that heme augmentation can cause a fast-acting lethal event for the parasite cells (advocated by findings of Timothy Egan and Paul D. Roepke). However, mefloquine is not a fast-acting drug, rather than, it is a relatively slow (slower than CQ, faster than atovaquone). Authors are encouraged to discuss that for mefloquine, most precedent literature of phenotype activity (10.1038/nmicrobiol.2017.31; 10.1126/scitranslmed.aau3174; 10.1021/acs.accounts.1c00154) point out that other mechanisms are operative rather than hemozoin blockage alone. Perhaps, mefloquine ability to inhibit hemozoin crystals is a secondary mechanism.

F) In the experimental design in Figure 5, it is not clear if a drug expose of 3 h was performed as denoted (3 or 6 hours). Indeed, a 3h data is only displayed in the supporting information. Otherwise, just kept 6 h in panel A.

G) In Panel E, of Figure 1, it was not clear in the "no drug" group what is the difference between the two columns? Is it 3-days versus 13 days?

H) Authors are encouraged to display, either in the main text or in the supporting information, a table with the full set of IC₅₀ values and standard deviation, in addition to the calculated fold change/ratio.

We thank the three reviewers for their support of our main findings and for the numerous helpful comments aimed at highlighting the context of our discoveries, improving the clarity of the text, and enhancing the potential impact of our results. The revisions introduced in response to their comments and suggestions have greatly improved the clarity of the presentation and the validity of the arguments.

Below we provide detailed accounts of the responses to the reviewers' comments.

Reviewer 1.

We thank Reviewer 1 for stating that our paper presents “an important contribution to the understanding of inhibition of β -hematin beyond classical mechanism based on presence of drug” and for classifying our results as interesting and with a potential to “open a way to future rational designs.”

Concern 1. Beyond the difference found in AFM experiment between PY, MQ or CQ with H-ART, what structural aspects could be dominant in the discrete inhibition of H-ART after drug removal?

Response 1. The mechanism of irreversible inhibition by H-ART initiates with the enhanced nucleation of nanocrystals in the presence of H-ART. The newly nucleated crystallites associate to the surface of larger growing crystals where they incorporate and strain the lattice. The generated lattice strain lowers the crystallization driving force and suppresses crystal growth. The sequence of events that follow enhanced crystal nucleation appears to be general, and are likely to be triggered by any compound that enhances nucleation. This expectation is supported by results with H-ARS. Thus, the question of the uniqueness of the mechanism of irreversible inhibition by H-ART transforms into why does H-ART enhance nucleation. We fully agree with Reviewer 1 that this is an extremely intriguing question on its own right. We are currently finalizing a study of how solution properties and mesoscopic solution aggregates couple to hematin crystal nucleation to address this point. This study will be published separately. The short answer is that H-ART and H-ARS, analogously to hematin, carry carboxyl groups. We use mixed aqueous-organic solvent to mimic the hemozoin environment in the parasite digestive vacuole. In this solvent the carboxyl groups of the heme adducts and hematin partially deprotonate and the resulting Coulomb repulsion between the carboxylate anions boosts the crystallization driving force and hematin crystal nucleation. Inhibitors PY, CQ, and MQ carry amino groups and do not have this effect.

Concern 2. Have author mediated the possibility to reply the biological studies by using CQ-resistant strain? It can be very interesting in order to explore the occurrence or nor of this non-classical mechanism in CQ-resistant strain of *P. falciparum*?

Response 1. We fully agree with Reviewer 1 that the responses to short drug pulses of a CQ-resistant strain would provide valuable insights. We have ongoing studies, to be published shortly, with the *P. falciparum* strain CAMWT, which is chloroquine resistant and CAM 580Y, which is isogenic and artemisinin ring-stage resistant. We would like to also refer to the work of Roepe that brought up the perspective that chloroquine resistance by PfCRT altered the continuous IC₅₀ but both were susceptible to high toxic doses of chloroquine. NF54 does not have isogenic CQ resistant isolate which would be better for comparison. Roepe also argues

that mutant PfCRT confers nearly all CQ cytostatic resistance as defined by an IC_{50} shift, but a much smaller component of CQ cytotoxic resistance as defined by a lethal dose (LD_{50}) shift. In this study, we are pulsing for short times with higher doses which might impose cytosolic stress in addition to more specific DV heme crystal inhibition by chloroquine.

Concern 3. In survival (%) vs. concentration plots of Figure 5, the units in x-axis are expressed as nm. Should be it mM?

Response 1. We thank Reviewer 1 for noticing that the notation for nanomolar was mistyped. We confirm that nM is correct. The drugs are effective at very lower concentrations.

Reviewer 2.

We thank Reviewer 2 for stating that the topic is quite important and the team of authors are prominent and highly skilled researchers.

Major issues:

Concern 1. The key data are presented in Fig. 1, and distinguishing whether blue and orange bars in Fig. 1e are the same or different in height is essential to interpretation, however, no statistics are given (also bars in extended data Fig. 2). From the error bars (extended Fig. 2 has no error bars at all) it is difficult to ascertain which blue and orange bars are statistically the same vs different. Routine T test and recitation of calculated p values is sorely needed.

Response 1. We thank Reviewer 2 for this valuable suggestion. We have carried out one-way ANOVA (equivalent to the t-test) of the similarities between the distributions of the crystal length and width increments. The ANOVA parameters and the suggested reversibility or irreversibility of inhibition are listed in the updated Supplementary Table 1 and extensively referenced in the text of the revised manuscript.

Concern 2. The authors propose that inhibition of crystal growth is irreversible if two criteria are met, the first being less crystal growth vs control (which is no inhibitor for 13 days) in the 10 days after inhibitor is removed following a 3 day incubation with inhibitor (criterion 1) and, second, that growth in the constant presence of inhibitor for 13 days is the same as growth following 3 days + followed by 10 days - inhibitor. Only 1 inhibitor (H-ARS) seems to satisfy both, but the title and text of the paper seems to imply that the inhibitors are irreversible, and Fig. 2 presents detailed arguments and a cartoon entitled "irreversible inhibition ... by H-ART" when the data in Fig. 1 show that H-ART satisfies neither criterion. The separation of crystal width vs length effects as reversible or irreversible in the text further confuses interpretation, are the authors implying that some inhibitors are reversible in one dimension but irreversible in another ? Some clarification in simple language is needed to assist the reader.

Response 2. We thank Reviewer 2 for pointing out that the discussion of the reversibility of inhibition of bulk crystallization may be confusing. We have rewritten this section of the paper relying on the ANOVA parameters—suggested in Concern 1—and we think that the revised version is substantially clearer. The revised discussion also emphasizes that, owing to the unique molecular structure of each crystal face, inhibitors are not expected to bind equally or to employ an identical mechanism to inhibit {100}, {010} and {011} faces of β -hematin crystals. We introduce the clarification “Notably the distinct structures of the anisotropic crystal faces select distinct modes of inhibitor binding (to the kinks or on the

terraces) and mechanisms and degrees of inhibition on each crystal face. Thus, we do not expect a drug to inhibit all faces uniformly reversibly or irreversibly.” Furthermore, we note that H-ART indeed does not inhibit both the {011} faces, which contribute to the crystal length, and the {010} faces, whose growth increases the crystal width; it inhibits irreversibly the {100} faces, as revealed by AFM observations of that face (Fig. 2). To address the concern that an irreversible inhibition of just one face may not have physiological consequences we introduced the following clarification in the revised manuscript: “even if a drug inhibits irreversibly only one of the hematin crystal faces (Fig. 1a), it will still delay the sequestration of hematin and contribute to the accumulation of this product of hemoglobin digestions. Thus, we expect the five compounds tested here to exhibit irreversible suppression of malaria parasites.”

Concern 3. The crystal inhibition assays are done over 13 days *in vitro* under highly non physiologic conditions, but the authors imply that by comparing growth effects of live parasites vs the same inhibitors that somehow the measured crystal inhibition characteristics are relevant to understanding the mechanism of drug inhibition of hemozoin *in vivo*, in which hemozoin crystals are formed within hours. The highly artificial nature of the crystal formation measurements needs to be emphasized, with limitations on interpretation then highlighted.

Response 3. In our *in vitro* assays we use biomimetic solvents with hematin concentrations similar to those in the parasite DV [Heller LE & Roepe PD (2018) *Biochemistry* 57(51):6927-6934]. Crystallization trials were carried out for extended times to test whether three distinct regimes of application of drugs and metabolites induced divergent crystal sizes. The comparisons between the average crystal sizes resulting from the three growth regimes indicated that mechanisms of irreversible inhibition of crystallization operate for the studied drugs and metabolites. These mechanisms were then confirmed in AFM experiments, which were carried out over times between one and three hours. Thus, the times of the bulk crystallization and the AFM tests bracket the times of parasite growth. We have modified the language in the revised manuscript to reflect the difference between death in 3 to 6 hours at the stage drug applied versus lingering drug in the digestive vacuole or on hemozoin, which persists to further damage the parasite growth after wash out. The off digestive vacuole target effects on ring stage cytosol toxicity versus trophozoite stage sensitivity may be different as more antioxidant molecules may be present, such as GSH, in the parasite cytosol at the trophozoite stage. We note that pyronaridine is more efficient at the ring stage killing than trophozoite, whereas chloroquine and amodiaquine were more efficient at killing at the trophozoite stage.

Additional points:

Concern 4. pg. 1 "inevitably predicts" but then the next sentence seems to contradict this.

Response 4. We thank Reviewer 2 for highlighting the possibility to misunderstand this paragraph. As Reviewer 2 correctly notes, there is a contradiction between the expectation based on reversibility of inhibitor adsorption and the observations of irreversibly stunted crystals. This contradiction is the main point of that statement. To highlight that we seek to emphasize the contradiction, we have rewritten the discussion to “The modifiers’ activities are commonly ascribed to their adsorption to specific crystal surface sites 18-21 and the reversibility of adsorption inevitably predicts that growth fully recovers after the inhibitor is removed 20,22. Multiple instances of permanently poisoned crystals 23 and terminal crystal sizes 24,25 contradict this prediction, stand out of the realm of the classical inhibition mechanisms, and have thus far remained elusive.”

Concern 5. next line, "irreversible inhibition of hemozoin crystallization" ... at best, "in vitro under non physiological conditions".

Response 5. We thank Reviewer 2 for highlighting the ambiguity of this statement. Debate continues on the exact heme crystal formation mechanism in parasite digestive vacuoles. The relevant statement in the revised manuscript now reads: "We explored the reversibility of inhibition of the growth β -hemozoin crystals (Fig. 1a), a synthetic analogue to hemozoin".

Concern 6. next line following, "cooperative", how so ? This term has a formal definition in biological sciences, it is not clear how cooperativity in inhibition of crystal growth by any inhibitor studied is being ascertained or quantified.

Response 6. Cooperativity describes a physical characteristic of many biological processes. One example is the binding of oxygen to hemoglobin, which is cooperative since the association of the first molecule of oxygen to one of the four hemoglobin hemes facilitates the attachment of three other oxygen molecules. The cooperativity of oxygen binding to hemoglobin is quantified by a Hill coefficient of 2.8. Crystal nucleation, including nucleation of hemozoin crystals, is also a cooperative process since the transition of one molecule into a new phase triggers the transition of many others. The Hill coefficient for crystal nucleation is infinity. Analogously, if a step on a crystal surface joins another step, this facilitates the accumulation of additional steps to the emerging step bunch. The cooperativity coefficient of step bunching varies with the properties of the crystal. In the revised version of the manuscript, we specify the latter two instances of cooperativity at several locations throughout the text. In the abstract, where we are constrained by the word limit, we specify that irreversible inhibition is enabled by distinct cooperativity mechanisms.

Concern 7. bottom of pg 1, top pg 2 is very misleading, H-ART and H-ARS used in the paper are not drugs, they are drug heme adducts. ART drugs cannot be "purged" from the solution as implied, they become covalently attached to their intracellular targets.

Response 7. We thank Reviewer 2 for highlighting the inaccurate use of the term "drugs". We have replaced it with several other terms, mostly "metabolites" and "compounds," but also, as appropriate, "agents" or "inhibitors." We fully agree that artemisinin-class drugs cannot be purged of the digestive vacuole and we only use the term to describe removal of the drugs from our crystallization container *in vitro*.

Concern 8. 5 lines following, "copious nucleation" is not defined or quantified, what is meant by this phrase ?

Response 8. Copious is frequently used in combination with nucleation to describe abundant nucleation of crystals without specifying the exact value of the nucleation rate.

Concern 9. pg 2 par 2 last line, reference 23 does not suggest "adducts ... form in the ... digestive vacuole" as implied, this paper uses NMR methods to assign meso carbon covalent attachment sites for ART - heme adducts formed *in vitro*.

Response 9. In the revised manuscript, we deleted reference 23 and introduced a reference to a paper from the Roepke's group, which demonstrates the formation of artemisinin adducts in parasites.

Concern 10. pg 12 last line second par, ref 32 measures killing rates which is not what is being measured here, but it is implied that what ref 32 and the authors measure is similar.

Response 10. The Sanz paper did use hypoxanthine to measure killing by dilution which was not done in our study. We reverted to the original Desjardins reference. Desjardins RE, Canfield CJ, Haynes JD, Chulay JD. Quantitative assessment of antimalarial activity in vitro by a semiautomated microdilution technique. *Antimicrobial Agents and Chemotherapy*. 1979;16:710–718.

Methods

Concern 11. "Materials"; were any of the chemicals or drugs purified or were they used as purchased ?

Response 11. We now specify “All materials were used as received”.

Concern 12. "Synthesis of Hematin" Sodium dithionite and artemisinin ... concentrations are not mentioned

Response 12. We thank Reviewer 2 for pointing out this crucial missing detail. The relevant statement now reads “Sodium dithionite at ca. 1 mM and artemisinin (ART) at ca. 1 mM were dissolved in DI water and *n*-butanol, respectively”.

Concern 13. "In Situ Monitoring ..." Please describe "... the liquid cell ..." is this a commercial or fabricated device, etc.

Response 13. We mean the standard liquid cell supplied with the microscope.

Concern 14. Top pg. 3 *how* was the solution in the fluid cell "...exchanged ..." and is the fluid cell the same as the liquid cell mentioned earlier ?

Response 14. We thank Reviewer 2 for bringing up the use of two terms for the same cell. Fluid cell is indeed the same as liquid cell. We have replaced the one instance of fluid cell in the Methods section with liquid cell. The standard AFM liquid cell has inlet and outlet ports that make solution exchange a routine procedure.

Concern 15. "Tests for Reversibility ..." 3rd line, H-ART and H-ARS are not drugs. Next line, why 2uM and 5uM for CQ and MQ ? Without explanation this seems arbitrary.

Response 15. We thank Reviewer 2 for noticing the inaccurate use of the term drugs. We have replaced all instances of drugs in the SI Methods with “additives”, “compounds”, or “inhibitors”.

Concern 16. pg 5 last par, drug and inhibitor concentrations are not listed. Also, "... parasite survival" is not the inverse of percent growth inhibition, but the inverse of parasite growth?

Response 16. We deleted this last sentence, which did not add to the paper about inverse of percent growth inhibition.

Reviewer 3:

We thank Reviewer 3 for highlighting several of our findings as novelties.

Concern 1. The notion that in low concentration, hematin-artemisinin adducts (H-ARS/H-ART) did not efficiently inhibit the growth of young parasites (early or late rings) is a substantial phenotype shift and a novelty in comparison to the parental Artemisinin/Artesunate efficacy. Presumably, adducts are devoid in peroxide bond necessary to alkylate protein; however, adducts were previously able, at least in a high concentration (500 nM), to kill early rings in the RSA (DOI 10.1074/jbc.RA120.016115). To reconcile this apparent paradox, authors are encouraged to discuss this or experimentally address the parasite survival in Figure 5 panels B and C using high concentrations of H-ARS/H-ART (up to 1 microM).

Response 1. A typical pulse drug assay for the artemisinin uses approximately 500 to 700 nM parent drug, which mirrors the pharmacokinetics of parent drugs as the basis for the ring stage survival assay. Here we explored lower concentration pulses of H-ART and H-ARS which is 5- to 50-fold less than those used in our previous JBC paper [Ma *et al.* (2021) JBC 296:100123], which Reviewer 3 has mentioned, to help differentiate stage specific actions. The intention was not to kill most of the rings seen with H-ARS with NF-54 and most of rings with H-ART, but to compare the stage dependence at lower drug concentrations. The 500 nM for instance showed near zero percent survival in the 6-hour pulse dose in our previous paper [Ma *et al.* (2021) JBC 296:100123].

Concern 2. Yet regarding the drug concentration, the authors report IC₅₀ values of 6 h expose versus 72 h (no wash out). The determination of IC₅₀s 6h was presumably intended to reflect the irreversibility of parasite inhibition and to precisely correlate this phenomenon with the drug interaction with hemozoin crystals; this is a novelty. A cut-off of IC₅₀ 6 h > 12 folds the IC₅₀ 72 h was wisely established. Troublingly, the curves of IC₅₀ values of 6 h expose for early and later ring do not seem like a sigmoid curve. The accuracy of regression-derived values is not clear either. In other words, why authors did not test compounds in higher concentration up to 1 microM in order to generate suitable sigmoid curves?

Response 2. We agree that the slope change for young rings with the heme-adduct are interesting for changes in the IC₇₅ or IC₉₀; however, we chose to concentrate on the IC₅₀. Inhibitor concentrations of 500 nM and 1000 nM do have more complete inhibition. We plan to explore pulsed application of higher doses of parasite suppressors in our future work and thank Reviewer 3 for this valuable suggestion.

Concern 3. In parts, a great novelty of the present study is the determination of phenotype signature of hematin-artemisinin adducts (H-ARS/H-ART). Their antiplasmodium activity are quite appealing. IC₅₀ values determined within 72 h of continuous drug incubation indicate they are almost equipotent, this is consistent to the structure-activity relationship. That said, there are dissimilarities in the IC₅₀ 6-h that should be addressed/discussed. There are a couple concerns with this experiment, though. First, H-ART seems to kill trophozoites more efficiently than H-ARS. Even in higher concentration, there are still parasites surviving at H-ARS treatment. Subsequently, trophozoite survival fractions for H-ART is of 5, while of 11 for H-ARS. Does this behave in the same way for IC₅₀ 3-h (Extended Data Fig. 5 not depicted for H-ARS)? Conversely, we know that iron protoporphyrin IX (Fe-PPIX) can adsorb in wire glass and the plastic surface of a microplate. No evidence is provided to indicate that the H-ARS/H-ART can be truly washed out by the protocol used. Therefore, it is possible that parasites continue to be effectively exposed to the hematin-artemisinin adducts following the washout step, especially if no plate transfer was performed (see plate transfer, DOI:

10.1128/AAC.00574-16). This could be the reason for the dissimilarity in IC₅₀ 6h values. This could be a useful feature in therapeutic applications, but confounds interpretation of phenotype response.

Response 3. During the pulse experiments, the parasites were held in Eppendorf tubes to allow for more efficient washes than transfer to fresh plastic 96 well plates. The Eppendorf tubes would be free from adherent heme-adduct. We are careful to note in the revised manuscript that we wash from an extracellular medium, but retained drug in the parasite or on hemozoin was demonstrated for the adducts in our previous paper [Ma *et al.* (2021) JBC 296:100123].

Concern 4. In a close inspection of the IC₅₀ values from Extended Data Fig. 6, all drugs except for mefloquine were quite consistent to the literature. For mefloquine, the IC₅₀ of 88 nM is higher than typically observed in most the literature. Mefloquine supplied by Sigma-Aldrich (M2319) is provided as a partially DMSO-insoluble salt. Could this be the reason for the limited potency of mefloquine?

Response 4. Mefloquine was used from powder Sigma#2319. Mefloquine is extremely lipophilic compared to the other quinolines used in our study. We did dissolve the molecule in DMSO before adding to the growth media. We agree that the IC₅₀ in the literature is closer to 40 nM for 3D7 and agree also that NF54 was the parent strain which is more gametocyte competent for mosquitoes than the clonal 3D7.

Concern 5. Yet regarding mefloquine. The irreversibility of parasite inhibition and the correlation with the drug interaction with hemozoin crystals (reversible/irreversible) is an important issue. For sure, all drugs tested here apart mefloquine (chloroquine, pyronaridine, and hemozoin-adducts) are of fast-action antimalarial activity (i.e., to decrease parasite viability over 24 h drug expose). Presumably, a fast-action property may correlate with the ability of a drug in inhibiting hemozoin crystal. It is largely assumed that heme augmentation can cause a fast-acting lethal event for the parasite cells (advocated by findings of Timothy Egan and Paul D. Roepe). However, mefloquine is not a fast-acting drug, rather than, it is a relatively slow (slower than CQ, faster than atovaquone). Authors are encouraged to discuss that for mefloquine, most precedent literature of phenotype activity (10.1038/nmicrobiol.2017.31; 10.1126/scitranslmed.aau3174; 10.1021/acs.accounts.1c00154) point out that other mechanisms are operative rather than hemozoin blockage alone. Perhaps, mefloquine ability to inhibit hemozoin crystals is a secondary mechanism.

Response 5. We agree that mefloquine may have many potential off hemozoin target effects especially with its lipid solubility, action on K⁺-channels and action noted in the cited papers. In the revised manuscript, we added “MQ ability to inhibit hemozoin crystals maybe a secondary mechanism to protein synthesis or *P. falciparum* purine nucleoside phosphorylase inhibition 37-39.” We thank Reviewer 3 for this suggestion.

Concern 6. In the experimental design in Figure 5, it is not clear if a drug expose of 3 h was performed as denoted (3 or 6 hours). Indeed, a 3h data is only displayed in the supporting information. Otherwise, just kept 6 h in panel A.

Response 6. We thank Reviewer 3 for noticing this potential misunderstanding and changed to time on the figure to 6 hours

Concern 7. In Panel E, of Figure 1, it was not clear in the “no drug” group what is the difference between the two columns? Is it 3-days versus 13 days?

Response 7. We thank Reviewer 3 for highlighting the lack of clarity. In the revised Fig. 1 the gray bars are clearly labeled.

Concern 8. Authors are encouraged to display, either in the main text or in the supporting information, a table with the full set of IC₅₀ values and standard deviation, in addition to the calculated fold change/ratio.

Response 8. We include the data for IC₅₀ for the 6- and 3-hour pulses in the revised Supplementary Table 2 and thank Reviewer 3 for this suggestion.

To summarize, we thank the reviewers again for their evaluation of our manuscript. We are confident that the editorial changes have adequately addressed the reviewers' comments.

REVIEWERS' COMMENTS:

Reviewer #1 (Remarks to the Author):

Dear Authors,

The present manuscript described a Nonclassical mechanisms to irreversibly suppress β -hematin crystal growth. The title is appropriate for the manuscript, solid evidence of the mechanism was shown, which was statistically well performed after revision. The conclusion are in good concordance with experimental results. The work is well written and well described. Novel and original data is presented, which can be of great importance not only to understand the potential irreversibly suppression of b-hematin crystallization, but that also, can represent a starting point to face the design of antimalarials based on 4-aminoquinolines from other perspective. I think that the information is valuable in this sense.

Authors made a strong effort to direct all reviewer queries and the revised version look like more clear.

Then, my report for the current manuscript is accept under current form.

Without further comments,

Sincerely yours,

Reviewer 1

Reviewer #2 (Remarks to the Author):

The authors have largely responded adequately to the previous critique, the following points and corrections should be addressed in the final manuscript:

Pg. 2

"...To model how drugs with limited residence times clear *P. falciparum* parasites, we probe whether antimalarials may adopt pathways that lead to inhibition of hematin crystal growth that lasts after an inhibitor has been removed from the system..."

Again, artemisinin drugs covalently attach to targets so they cannot be "... removed from the system ..." ("system" having the normally assumed meaning of the solution, the infected red cell, and the parasite). Non activated or unreacted artemisinin drugs can be removed from the surrounding medium, ("the surroundings"). Some clarification is needed so readers unfamiliar with Art drug reduction – oxidation activation and subsequent target alkylation chemistry are not confused. These are essential aspects to how the drugs work and should not be ignored or portrayed incorrectly.

Same page, further down

"...To promote the physiological relevance of the obtained results..."

Suggest "To test" the physiological relevance of the obtained results

Pg. 15, The statement:

"... Furthermore, all five compounds partially protonate upon invading the digestive vacuole to adjust to its pH (ca. 5.0 36) lower than that of blood and the erythrocyte cytosol (ca. 7.35), but undergo no further chemical modifications ..."

Again, is incorrect, ART drugs are reduced and form oxygen and carbon centered radicals when they are reduced within the parasite, even for rings. If the authors believe that some portion of Art based drugs they introduced remain in their oxidized, non reactive, diffusible form after being introduced into the parasite culture, they can examine this by a number of assays, else all other data suggest that ART drugs are indeed "chemically [modified]", even in early ring stages.

Pg. 16

"...This pulse metabolite or drug assay differs from the artemisinin ring stage pulse assays that look at artemisinin drugs on ring stages from genetically diverse malaria parasites...."

Suggest "examine" instead of "look at", further, perhaps the authors mean "examine the effects of" ?

"...The newly identified correlation between irreversible crystallization inhibition and irreversible suppression of malaria parasites suggests that irreversible inhibition of hemozoin crystallization may be essential for the antimalarial/ antiparasitic activity. ..."

And elsewhere,

Suggest change "correlation" to "relationship", throughout, there is no correlation shown in the formal mathematical sense.

Reviewer #3 (Remarks to the Author):

The revised version of the manuscript Nonclassical mechanisms to irreversibly suppress β -hemozoin crystal growth authored by Wenchuan Ma has edited as well as addressed most of my suggestions.

Reviewer 1.

We thank Reviewer 1 for stating that the title is appropriate for the manuscript, solid evidence of the mechanism was shown, which was statistically well performed after revision. The conclusions are in good concordance with experimental results. The work is well written and well described. Novel and original data is presented, which can be of great importance not only to understand the potential irreversibly suppression of β -hematin crystallization, but that also, can represent a starting point to face the design of antimalarials based on 4-aminoquinolines from other perspective. I think that the information is valuable in this sense.

Reviewer 1 also acknowledges that the authors made a strong effort to direct all reviewer queries and the revised version look like more clear.

Reviewer 1 has no critical comments or suggestions on this version of the manuscript.

Reviewer 2.

We thank Reviewer 2 for stating that the authors have largely responded adequately to the previous critique.

Concern 1. , artemisinin drugs covalently attach to targets so they cannot be “... removed from the system ...” (“system” having the normally assumed meaning of the solution, the infected red cell, and the parasite). Non activated or unreacted artemisinin drugs can be removed from the surrounding medium, (“the surroundings”). Some clarification is needed so readers unfamiliar with Art drug reduction – oxidation activation and subsequent target alkylation chemistry are not confused. These are essential aspects to how the drugs work and should not be ignored or portrayed incorrectly.

Response 1. We thank Reviewer 2 for this suggestion. We have modified the text to “Several antimalarial compounds, such as quinoline-class antimalarials ^{3,4} and the hematin adducts of artemisinin class drugs, produced by the parasite metabolism ^{5,6}, kill the parasites by inhibiting hematin crystallization, which boosts the concentration of toxic free hematin.”

Concern 2. Suggest "To test" the physiological relevance of the obtained results

Response 2. Reviewer 2 refers to a statement in the Introduction, in which we discuss the solvent used in our *in vitro* studies. This solvent consists of octanol saturated with citric buffer, designed to mimic the composition of the lipid nanospheres in the parasite digestive vacuole, as analyzed in the literature. We rely on published results on how appropriate the use of this solvent as a model of the *in vivo* lipids is. Since we do not test how well this solvent mimics the composition of the lipid nanodroplets, it appears to us that the use of “promote” instead of “test” better reflects the purpose of using this solvent.

Concern 3. The statement “... Furthermore, all five compounds partially protonate upon invading the digestive vacuole to adjust to its pH (ca. 5.0 ³⁶) lower than that of blood and the erythrocyte cytosol (ca. 7.35), but undergo no further chemical modifications ...”

Again, is incorrect, ART drugs are reduced and form oxygen and carbon centered radicals when they are reduced within the parasite, even for rings. If the authors believe that some portion of

Art based drugs they introduced remain in their oxidized, non reactive, diffusible form after being introduced into the parasite culture, they can examine this by a number of assays, else all other data suggest that ART drugs are indeed "chemically [modified]", even in early ring stages.

Response 3. The five compounds that we discuss in the statement highlighted by Reviewer 2 include three quinoline-class antimalarials and the heme adducts of artemisinin and artesunate. In contrast to the drugs artemisinin and artesunate, the respective heme adducts, which represent the final stages of their metabolisms, have not been shown to undergo any further chemical modification in the parasite DV.

Concern 4. Suggest "examine" instead of "look at", further, perhaps the authors mean "examine the effects of"

Response 4. We thank Reviewer 2 for this suggestion, which we gladly adopt in the newly revised version.

Concern 5. Suggest change "correlation" to "relationship", throughout, there is no correlation shown in the formal mathematical sense.

Response 5. We thank Reviewer 2 for highlighting that "correlation" may have been used out of its prime meaning. We have replaced it in page 16 of the newly revised version with "correspondence". Throughout the text, "correlation" is used to indicate either true mathematical relations between two variables, or the light scattering correlation function. We have left these instances of "correlation" use unchanged

Reviewer 3:

We thank Reviewer 3 for stating that the revised version of the manuscript Nonclassical mechanisms to irreversibly suppress β -hematin crystal growth authored by Wenchuan Ma has edited as well as addressed most of my suggestions.

Reviewer 3 has no critical comments or suggestions on this version of the manuscript.